# Attention Mechanism-Based Glaucoma Classification Model Using Retinal Fundus Images

**DOI:** 10.3390/s24144684

**Published:** 2024-07-19

**Authors:** You-Sang Cho, Ho-Jung Song, Ju-Hyuck Han, Yong-Suk Kim

**Affiliations:** 1Department of Biomedical Engineering, Konyang University, Daejeon 35365, Republic of Korea; 24856504@konyang.ac.kr (Y.-S.C.); songhj6692@gmail.com (H.-J.S.); 21856503@konyang.ac.kr (J.-H.H.); 2Department of Artificial Intelligence, Konyang University, Daejeon 35365, Republic of Korea

**Keywords:** attention, classification, causality, artificial intelligence

## Abstract

This paper presents a classification model for eye diseases utilizing attention mechanisms to learn features from fundus images and structures. The study focuses on diagnosing glaucoma by extracting retinal vessels and the optic disc from fundus images using a ResU-Net-based segmentation model and Hough Circle Transform, respectively. The extracted structures and preprocessed images were inputted into a CNN-based multi-input model for training. Comparative evaluations demonstrated that our model outperformed other research models in classifying glaucoma, even with a smaller dataset. Ablation studies confirmed that using attention mechanisms to learn fundus structures significantly enhanced performance. The study also highlighted the challenges in normal case classification due to potential feature degradation during structure extraction. Future research will focus on incorporating additional fundus structures such as the macula, refining extraction algorithms, and expanding the types of classified eye diseases.

## 1. Introduction

Age-related macular degeneration, glaucoma, and diabetic retinopathy are the three leading causes of blindness, with higher prevalence rates in older populations. Despite their high prevalence, the awareness of these eye diseases is very low. The damage caused by these major blinding diseases progresses slowly, is irreversible, and leads to visual field defects [1,2]. Therefore, the early detection of these diseases through fundus imaging before the damage progresses is crucial.

Fundus images provide clinical information obtained from fundus examinations used to measure and diagnose the condition of the eye. These non-invasive images allow for the observation of blood vessels within the body and are widely used in clinical practice for various diseases [3]. Additionally, as shown in Figure 1, fundus imaging allows the observation of structures such as retinal vessels, the optic disc, and the macula, where the optic nerve fibers converge and enter the brain. Among these structures, the optic disc refers to the location where the optic nerve, formed by the convergence of nerve fibers originating from the retina, exits the back of the eye. Visual cells convert light stimuli into signals that can be understood, and these received stimuli are transmitted to numerous ganglion cells branching from the visual nerve. These ganglion cells then gather these stimuli and send them to the optic disc. From the optic disc, neural transmission proceeds to the brain [4]. It is also a crucial point where major retinal blood vessels enter the fundus. As shown in Figure 1, glaucoma is characterized by an increased cup-to-disc ratio. The increase in intraocular pressure due to these characteristics can also lead to damage to the optic nerve, which is especially problematic in the case of glaucoma. Glaucoma is often asymptomatic in its early stages, but the increased intraocular pressure slowly damages the optic nerve, leading to irreversible vision loss and, if untreated, blindness. This silent progression makes regular screening and early detection essential for managing the disease effectively. Therefore, monitoring the optic disc and other fundus structures is vital for the timely diagnosis and treatment of glaucoma.

Thus, fundus structures such as the optic disc play a vital role in the diagnosis of glaucoma. However, interpreting fundus images requires skilled ophthalmologists, and there are more patients than there are specialists, leading to a shortage of medical personnel. To address these issues, research on artificial intelligence and machine learning techniques for diagnosing eye diseases using fundus images is actively being conducted.

To identify glaucoma, Orlando et al. implemented two CNN transfer learning methods, OverFeat and VGG-S, using vascular in-painting and CLAHE (Contrast Limited Adaptive Histogram Equalization) techniques for training [5]. Sudhan, M. B. et al. proposed a DCNN (Deep Convolution Neural Network) model that extracts the optic disc from fundus images and then uses these images to diagnose glaucoma [6]. Additionally, research is being conducted to diagnose various diseases from fundus images [7,8,9,10]. Dipu et al. studied the classification performance of CNN-based models such as ResNet-34, EfficientNet, MobileNetV2, and VGG-16 in computer vision for fundus image classification [7]. Tareq Babaqi et al. proposed a pretrained CNN model based on transfer learning, and Demir et al. suggested an approach using sequence models [8,9]. Gour Neha and Pritee Khanna proposed a multi-input model that diagnoses eye diseases by inputting two fundus images, one from the right eye and one from the left eye, into the model [11].

However, the existing studies do not reflect the information used in actual clinical settings. They do not incorporate the unique features of fundus structures that ophthalmologists use to identify different eye diseases. Additionally, most prior research primarily involves training models by inputting the entire fundus image. Conversely, even when learning is conducted on fundus structures, it is not performed with multiple inputs, thus not reflecting the entire image. As a result, the relationship between the model’s output and the overall image cannot be explained.

Figure 2 is the overall flow of the method proposed by this paper. To address these issues, this paper extracts retinal vessels and the optic disc from fundus images and inputs them individually into the AI model. The retinal vessels are extracted using a segmentation model based on ResU-Net, while the optic disc is extracted using an algorithm based on the Hough Circle Transform [12]. The extracted data are then processed using an attention mechanism [13] to learn the preprocessed fundus images and structural features according to the eye disease, thereby incorporating clinical information. This paper describes the preprocessing process applied to the dataset after introducing the dataset. After explaining the proposed glaucoma classification model, the experimental results of the model are described in the order of discovery.

## 2. Materials and Methods

### 2.1. Dataset

The data used in this paper are from the ODIR (Ocular Disease Intelligent Recognition) dataset available on Kaggle [14]. The ODIR dataset is a database of fundus images of the left and right eyes, labeled with diagnoses made by ophthalmologists. These data was collected from actual patients at hospitals and medical centers in China by Shanggong Medical Technology. In these institutions, fundus images are captured by various cameras available in the market, such as Canon, Zeiss, and Kowa, resulting in varied image resolutions. All fundus images used in this study are standardized to 512 × 512 in size and consist of three channels. The labels include eight categories: normal, myopia, hypertension, diabetic retinopathy, cataract, glaucoma, age-related macular degeneration, and other diseases. The dataset also provided annotations that were labeled by trained human readers with quality control management. The composition of the training data is shown in Table 1. In this paper, we used two labels from the ODIR dataset: glaucoma and normal. We used the fundus images of the right eye and excluded low-quality images affected by glare or other issues. To match the number of glaucoma and normal data, 32 samples of glaucoma and 32 normal data were selected for each label.

To effectively extract fundus structures and classify eye diseases, we preprocessed the ODIR data. Figure 3 shows the preprocessing process used in this paper: (a) the original ODIR data. As shown in (a), to extract the background area, which is the black portion excluding the actual eye region in the fundus image, we converted the data from the RGB channel to the gray channel. Then, we used the thresholding technique to binarize the image based on a specific threshold to extract the black background. During the thresholding process, we set the threshold at 40 within the grayscale range of 0 to 255. Pixels with brightness values above 40 were converted to white, distinguishing the eye region from the black background in the fundus image. The black background has a high contrast with the eye region, which can damage image features during histogram equalization. Therefore, as shown in (b), the background color was adjusted to match the average RGB channel values of each fundus image to resemble the eye color as closely as possible. To apply histogram equalization only to the brightness channel, the images were converted to the Lab channel, as shown in (c). CLAHE was then used to enhance the contrast of the fundus image boundaries, and the images were converted back to the RGB channel [15]. Images (d) and (e) show the images after CLAHE was applied to each brightness channel and then converted back to the RGB channel. Finally, in Figure 3f, the original black background was reapplied to the image converted to RGB, making the preprocessing similar to the original image composition.

### 2.2. Process of Data Collection and Preprocessing of Fundus Structures

The 32 preprocessed images were divided into 30 for training data and 2 for validation data to prevent overfitting during model training. To augment the number of images, they were rotated by 10 degrees each time. The data input to the glaucoma classification model was resized to 64 × 64. Each label had 1080 training images and 72 validation images after augmentation.

The segmentation model used for retinal vessel extraction in this paper is ResU-Net, which is built on the U-Net architecture [16]. As shown in Figure 4, it is composed of six Residual Blocks. ResU-Net employs a concatenation method in the encoding stage, combining the features obtained from each layer with the corresponding layer in the decoding stage. This direct connection between the encoder and decoder layers is referred to as a Skip Connection. The parts connecting the convolution blocks in the middle of the encoder and decoder, as shown in Figure 4, are called Bridges. To effectively extract retinal vessels, the preprocessed images were converted to the gray channel and resized to 256 × 256. Each image was then divided into sixteen 64 × 64 segments and input into ResU-Net to extract the retinal vessels. The extracted retinal vessels were recombined into the original 256 × 256 size and then resized to 64 × 64 for use in this study. Figure 4 shows the structure of the ResU-Net used and the retinal vessel images extracted. (a) depicts the structure of the ResU-Net utilized, and (b) shows the retinal vessel image extracted from the fundus image using the model in (a).

Since only the right eye’s fundus image was utilized, the image was split in half to extract only the right portion for extraction. Figure 5 illustrates a brief process of optic disc extraction. Similarly to retinal vessel extraction, the optic disc extraction process began with conversion to the Lab channel for ease of preprocessing from the original data with similar colors. Subsequently, for optic disc detection, as it typically exhibits brighter intensity within the fundus image, the top 3.5% of pixels in the brightness channel of the Lab channel were extracted to detect it. The extracted pixels were then utilized in the Hough Circle Transform to detect circles in the image, thereby identifying the optic disc [12]. To prevent the optic disc from being partially outside the circle’s range during detection, the radius of the circle was extended by a length of 40 when detecting circles. After applying the specified range of the optic disc to the original data, areas outside the range were converted to black to highlight the features of the optic disc, followed by resizing to 64 × 64.

### 2.3. Glaucoma Classification Model

The glaucoma classification model in this paper was constructed based on a CNN structure, as illustrated in Figure 6. It was designed as a multi-input model, simultaneously inputting the preprocessed images and the two previously extracted fundus structures: retinal vessels and the optic disc. The input images are 64 × 64 in size and composed of a single channel, fed into the CNN module for individual learning. The CNN module consists of four Convolution Layers, each followed by BatchNormalization and Maxpooling twice. All Convolution Layers have a filter size of 3 × 3, and the activation function is set to ReLU (Rectified Linear Unit). The number of filters gradually increases to 64, 128, 256, and 512.

After passing through two Convolution Layers, BatchNormalization and Maxpooling are applied, with the result after the final Maxpooling designated as the output. The results from the CNN module are then input into the Attention Layer. The attention results from retinal vessel and optic disc were concatenated using the Concatenate technique to combine their respective feature values. The combined feature values underwent normalization through BatchNormalization and Maxpooling. The normalized feature values were then input into Convolution Layers and normalized again to learn the feature values obtained from all input images. As shown in Figure 6, the number of filters for the Convolution Layers was set to 2048, 4096, and 4096, with the filters of the final Convolution Layer set to 1 × 1 to learn the overall features. To classify glaucoma, the learned feature values were processed through Fully Connected (FC) Layers and trained using Dense layers, with final activation performed using ReLU and Softmax.

## 3. Results

The performance of the classification model was evaluated using the Area Under the Curve–Receiver Operating Characteristic (AUC-ROC) curve. The ROC curve is a measurement graph that depicts the model’s classification performance at different thresholds. The ROC curve shows the model’s performance across all thresholds, and the area under the ROC curve is referred to as AUC. Higher AUCs indicate that the model is excellent at distinguishing labels, and the proposed model received 0.91 points, which can be seen in Figure 7. Table 2 presents performance metrics of the classification model. The performance of disease classification was evaluated in terms of sensitivity, precision, and F1-Score. Glaucoma showed the highest performance across all three metrics, with values above 0.97, while the classification performance for normal cases also yielded values above 0.81.

To evaluate the model’s performance comprehensively, sensitivity, precision, and F1-Score were selected. Sensitivity measures the model’s ability to correctly identify true positives, which is crucial for detecting diseases like glaucoma. Precision evaluates the accuracy of the positive predictions, ensuring that the model does not produce excessive false positives. F1-Score balances the trade-off between sensitivity and precision, providing a single metric that considers both, particularly useful when dealing with imbalanced datasets. In Table 2, the performance of the classification model constructed in this paper is compared with models from comparative studies. Sensitivity, precision, and F1-Score for each label are compared. The results of the model with the highest performance are highlighted in bold, and the second-highest results are underlined. Regarding glaucoma, the model used in this paper exhibited the best performance in performance metrics except for sensitivity. It achieved a precision of 1.00 and an F1-Score of 0.99, showing a difference of 0.06 in precision compared to the second-best model. Additionally, its sensitivity was only 0.01 lower than the model with the highest performance (Abbas, Qaisar, et al., 2024 [10]). Such a performance is attributed to the results obtained based on the types of eye structures inputted. For glaucoma, as diagnosis primarily relies on the optic disc in fundus images, the model demonstrated higher classification performance compared to models in other studies.

Ablation studies were conducted to verify the effect of image attention, and the results are shown in Table 3 The model without attention is denoted as ‘Our model w/o Attention’. This model combines the weights output from the three CNN modules through concatenation instead of attention for training. The results showed that ‘Our model w/o Attention’ recorded a sensitivity, precision, and F1-score of 0.77, 0.83, and 0.80, respectively, for glaucoma, and 0.82, 0.75, and 0.78 for normal cases. This performance is inferior compared to the model with attention. The significant performance gap, especially in glaucoma, suggests that leveraging attention to learn features of fundus structures was indeed effective.

## 4. Discussion

Through performance comparison evaluations and results, the learning performance based on the input of fundus structures was confirmed. The classification performance for glaucoma, which is diagnosed through retinal vessels and the optic disc, recorded a superior performance compared to other research models. The classification performance for glaucoma was the best even with a smaller dataset compared to other models. However, the performance for classifying normal cases was lower than that of other research models. This is presumed to be due to some degradation of features during the extraction of fundus structures depending on the image. Additionally, it is likely that confusion occurred during learning due to the feature values of the retinal vessels when learning the feature values of the optic disc. Despite these limitations, the distinctiveness of our approach lies in its focused analysis of fundus structures to diagnose glaucoma, demonstrating the potential for high accuracy even with limited data. Our findings underline the importance of high-quality, standardized imaging and suggest that enhancements in image preprocessing and feature extraction algorithms could further improve classification performance. Future research will need to address these factors, potentially by employing more sophisticated models or integrating additional data types to bolster robustness and accuracy.

## 5. Conclusions

In this paper, to incorporate clinical information into the classification model, fundus images and fundus structures were learned by the AI model using the attention mechanism. We compared and evaluated the performance of the classification model proposed in this paper with models from other studies. Additionally, through experiments, including ablation studies, we confirmed that using attention to learn fundus structures enhances performance. For future research, we plan to improve classification performance by utilizing fundus structures that were not included in this study, such as the macula. Additionally, we will work on refining the algorithms used to extract fundus structures to enhance the clarity of these extractions, thereby improving the model’s performance. Finally, we aim to expand the types of eye diseases classified beyond glaucoma and augment the original dataset.

## Figures and Tables

**Figure 1 sensors-24-04684-f001:**
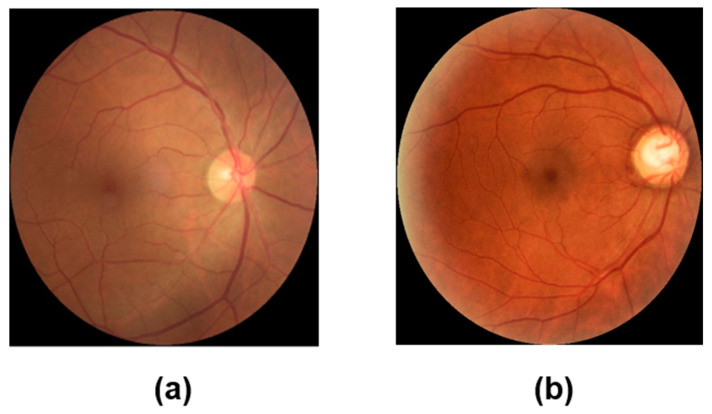
Examples of fundus images: (**a**) normal image; (**b**) fundus image of glaucoma patients.

**Figure 2 sensors-24-04684-f002:**
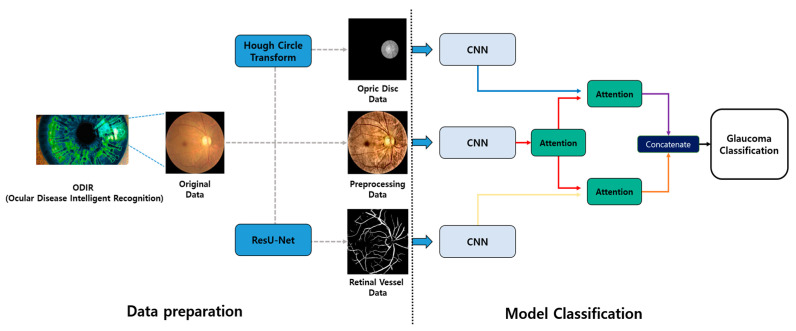
Overall flowchart of the method proposed in this paper.

**Figure 3 sensors-24-04684-f003:**
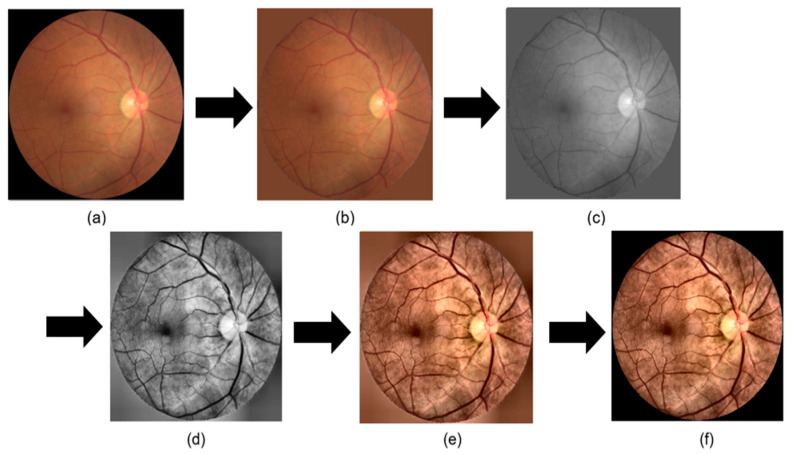
Process of preprocessing. (**a**) The original data used in this study; (**b**) the data applying the average RGB channel value of the image to the background area around the fundus; (**c**) the data using only the L channel after the change to the Lab channel; (**d**) the data applying CLAHE to (**c**); (**e**) the data changed to the RGB channel after the previous process; and (**f**) the data applying the existing black background in (**e**).

**Figure 4 sensors-24-04684-f004:**
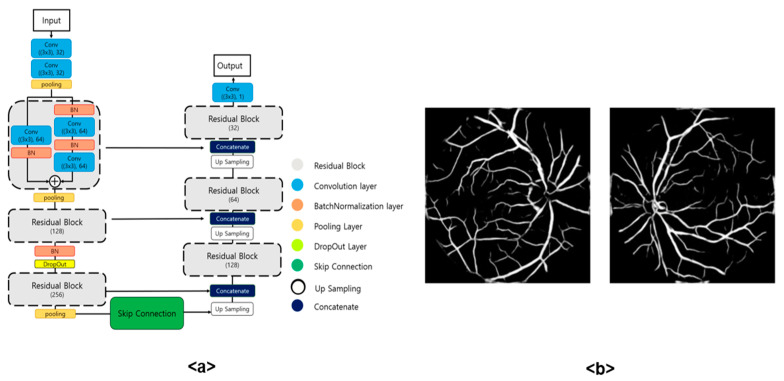
Architecture of Retinal Vascular Segmentation Model and extracted Retinal Vascular Data: (**a**) built on ResU-Net as architecture of retinal vascular segmentation model; (**b**) a retinal vascular image extracted through ResU-Net, where the left side is an image extracted from glaucoma data, and the right side an image extracted from normal data.

**Figure 5 sensors-24-04684-f005:**
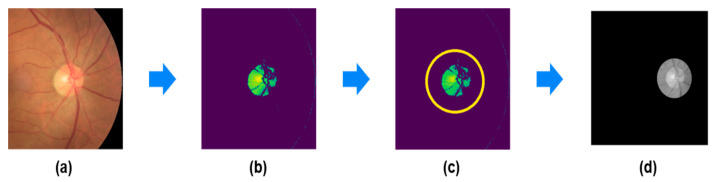
Optic nerve nipple extraction process. (**a**) Only the right part of the fundus image is extracted; (**b**) the top 3.5% of all pixels within the L channel after converting to the Lab channel; (**c**) using the Hough Circle Transform to detect circles in the image to specify the range; (**d**) applying a specific range to the fundus image.

**Figure 6 sensors-24-04684-f006:**
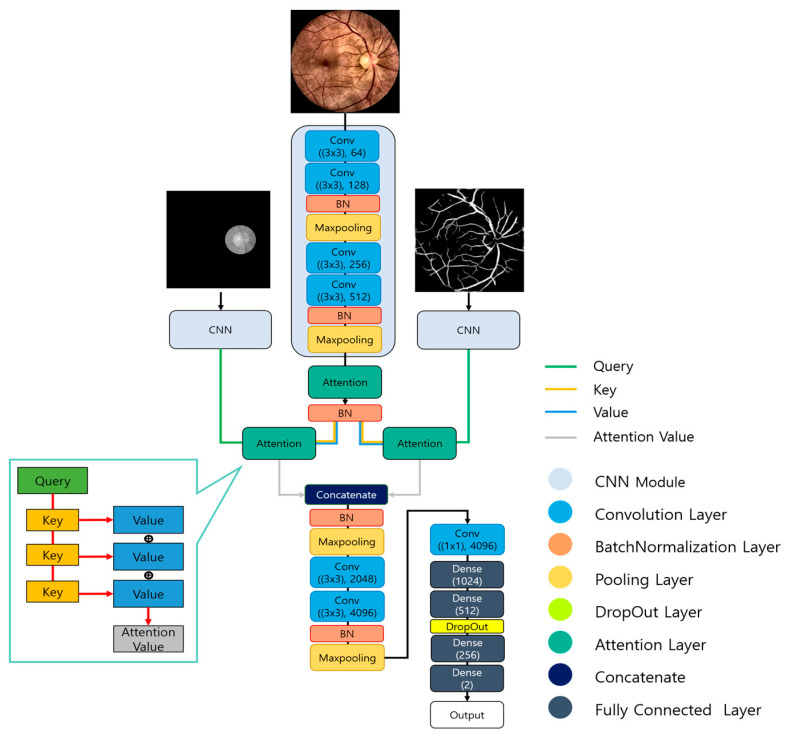
Architecture of the glaucoma classification model.

**Figure 7 sensors-24-04684-f007:**
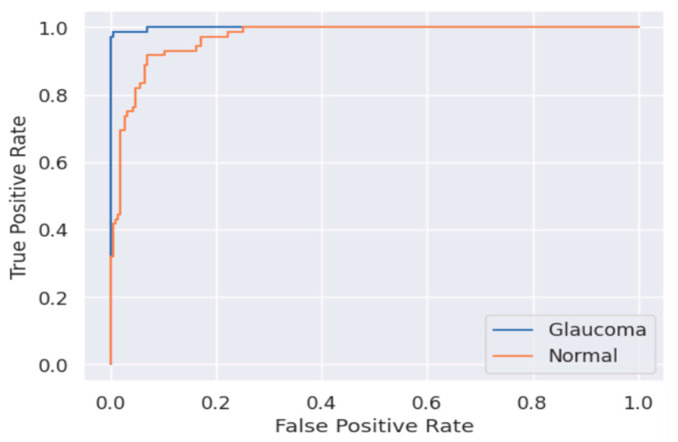
ROC curve of glaucoma classification model.

**Table 1 sensors-24-04684-t001:** Number of data for glaucoma and normal during ODIR dataset configuration.

Type of Disease	Number of Train Sets
Normal	1135
Glaucoma	207

**Table 2 sensors-24-04684-t002:** Comparison results with models of other studies that distinguish glaucoma.

	Model	Sensitivity	Precision	F1-Score
Glaucoma	VGG-16 [7]	0.95	0.92	0.93
	Transfer Learning + CNN [8]	0.85	0.94	0.89
	R-CNN + LSTM [9]	0.82	0.94	0.88
	AlexNet + ReliefF + XgBoost [10]	**0.98**	-	0.98
	Our Model	0.97	**1.00**	**0.99**
Normal	VGG-16 [7]	0.94	0.93	0.93
	Transfer Learning + CNN [8]	0.96	0.86	0.91
	R-CNN + LSTM [9]	0.97	0.84	0.90
	AlexNet + ReliefF + XgBoost [10]	**0.98**	-	**0.98**
	Our Model	0.86	0.81	0.83

**Table 3 sensors-24-04684-t003:** Ablation results of our model and its variants.

	Model	Sensitivity	Precision	F1-Score
Glaucoma	Our model w/o Attention.	0.77	0.83	0.80
	Our Model	0.97	1.00	0.99
Normal	Our model w/o Attention.	0.82	0.75	0.78
	Our Model	0.86	0.81	0.83

## Data Availability

This paper utilized the open dataset, the ODIR dataset, which can be accessed at the following link: https://www.kaggle.com/datasets/andrewmvd/ocular-disease-recognition-odir5k (accessed on 3 June 2024).

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
