# Peer review of "Attention Mechanism-Based Glaucoma Classification Model Using Retinal Fundus Images"

_sensors, 2024, doi:10.3390/s24144684_

Round 1

Reviewer 1 Report

Comments and Suggestions for Authors

In this manuscript, the authors proposed a novel attention based deep learning model to facilitate Glaucoma classification. Glaucoma is a highly common disease in Ophthalmology, and deep learning is most of the most popular technologies in medical imaging field. Thus this study is highly significant. Overall, the whole research is well organized and the manuscript is well written. Meanwhile, the reviewer has the following suggestions:

  1. The paper title is glaucoma classification, but the introduction mentioned the AMD, glaucoma, and diabetic retinopathy, please revise and add more introduction for glaucoma.

  2. The Table 1 contain many different categories, please remove the unrelated information. Meanwhile, please clarify the description “Each label consisted of 32 samples”. It is unlikely that only 32 sample is used for model training. 

  3. The dataset in this study is imbalance (207 vs 1135), please add the data augmentation information. If such a method is not used, please explain why.

  4. Please add ROC curve and confusion matrix in the results section, area under the curve values should also be provided.

  5. Discussion is too short, please add discuss more about the uniqueness/limitations of this study.

Author Response

1.The paper title is glaucoma classification, but the introduction mentioned the AMD, glaucoma, and diabetic retinopathy, please revise and add more introduction for glaucoma.

→ Added more information about glaucoma in the introduction: 'which is especially problematic in the case of glaucoma. Glaucoma is often asymptomatic in its early stages, but the increased intraocular pressure slowly damages the optic nerve, leading to irreversible vision loss and, if untreated, blindness. This silent progression makes regular screening and early detection essential for managing the disease effectively. Therefore, monitoring the optic disc and other fundus structures is vital for the timely diagnosis and treatment of glaucoma.'

2.The Table 1 contain many different categories, please remove the unrelated information. Meanwhile, please clarify the description “Each label consisted of 32 samples”. It is unlikely that only 32 sample is used for model training.

→ Added more detailed explanation: 'To match the number of glaucoma and normal data, 32 samples of glaucoma and 32 samples of normal data were selected for each label.'

3.The dataset in this study is imbalance (207 vs 1135), please add the data augmentation information. If such a method is not used, please explain why.

→ Instead of using all data from ODIR, as added in the sentence 'To match the number of glaucoma and normal data, 32 samples of glaucoma and 32 samples of normal data were selected for each label,' we utilized a total of 64 original images, with 32 for each label. After preprocessing, as added in the sentence 'The 32 preprocessed images were divided into 30 for training data and 2 for validation data to prevent overfitting during model training. To augment the number of images, they were rotated by 10 degrees each time. The data input to the glaucoma classification model was resized to 64 × 64. Each label had 1,080 training images and 72 validation images after augmentation.'

4.Please add ROC curve and confusion matrix in the results section, area under the curve values should also be provided.

→ Added the relevant figures to the results section and related sentences: 'Higher AUCs indicate that the model is excellent at distinguishing between labels, and the proposed model scored 0.91.'

5. Discussion is too short, please add discuss more about the uniqueness/limitations of this study.

→ Added to the discussion: 'Despite these limitations, the distinctiveness of our approach lies in its focused analysis of fundus structures to diagnose glaucoma, demonstrating the potential for high accuracy even with limited data. Our findings underline the importance of high-quality, standardized imaging and suggest that enhancements in image preprocessing and feature extraction algorithms could further improve classification performance. Future research will need to address these factors, potentially by employing more sophisticated models or integrating additional data types to bolster robustness and accuracy.'

Reviewer 2 Report

Comments and Suggestions for Authors

This paper presents a new model for classifying glaucoma images using artificial intelligence and machine learning techniques to extract retinal vessels and the optic disc from fundus images. The paper is interesting; however, several issues need to be addressed:

  1. It is important to clarify how the segmentation model and the classification model were selected and how they compare to previous models used in related works. This will help clarify the contribution of this paper.
  2. The methodology is not clear. I recommend adding a chart or figure that clarifies the steps, based on the scientific method.
  3. The architecture presented in figure 5 is very general, it needs to be defined more specifically.
  4. The results in Table 2 are not clear. The authors need to clarify how the metrics were obtained and justify the use of the selected metrics by comparing them with others that could have been chosen. Additionally, the table title is incorrect.
  5. The image quality needs to be improved.
Comments on the Quality of English Language

I recommend improving the titles in the tables

Author Response

It is important to clarify how the segmentation model and the classification model were selected and how they compare to previous models used in related works. This will help clarify the contribution of this paper.
→ For the selection of the segmentation model, research was conducted through academic conferences in Korea. This will be further refined and detailed in future research, so it has not been included in this paper. Additionally, regarding the classification model, this paper represents the first stage, and comparisons with other classification models will be made in future studies.

The methodology is not clear. I recommend adding a chart or figure that clarifies the steps, based on the scientific method.
→ The relevant figure has been added to the introduction.

The architecture presented in figure 5 is very general, it needs to be defined more specifically.
→ The figure has been updated to include more specific information related to the Attention mechanism.

The results in Table 2 are not clear. The authors need to clarify how the metrics were obtained and justify the use of the selected metrics by comparing them with others that could have been chosen. Additionally, the table title is incorrect.
→ "To evaluate the model's performance comprehensively, Sensitivity, Precision, and F1-Score were selected. Sensitivity measures the model's ability to correctly identify true positives, which is crucial for detecting diseases like glaucoma. Precision evaluates the accuracy of the positive predictions, ensuring that the model does not produce excessive false positives. F1-Score balances the trade-off between Sensitivity and Precision, providing a single metric that considers both, particularly useful when dealing with imbalanced datasets." was added to the Results section. Additionally, the title of Table 2 has been corrected.

The image quality needs to be improved.
→ The overall image quality has been improved.

Reviewer 3 Report

Comments and Suggestions for Authors

The paper discusses an interesting novel approach to classify fundus image and categorize them into normal vs. glaucoma.

While the results look impressive, the paper clearly seems to come from an informatics perspective and basic understanding of the underlying medical conditions seems to lack. This needs to be improved – preferably with the help of an ophthalmologist -, so that an ophthalmologist would see that the methodology and resulting mechanism of action are correct and that he could actually trust the data.

This lacking medical understanding is first of all that glaucoma is presented as a diagnosis that can be performed on a fundus photography image (lines 213, 229). This is simply wrong. Diagnosis of glaucoma usually includes measurement of intraocular pressure, visual field testing, OCT, and funduscopy, and more often that more than one examination for a time course and determination of progression is needed. Fundus photography could be used to differentiate between glaucoma-suspect vs. non-glaucoma-suspect optic disks, or maybe to differentiate between healthy vs. late-stage-glaucoma with severe optic disk impairment, but not generally between healthy vs. glaucoma.

Therefore, first of all a better description of the ODIR dataset is needed. How was the diagnosis glaucoma confirmed, and what were the criteria for healthy patients? Without further description, distinguishing between healthy vs. glaucoma solely based on fundus photography with a sensitivity/specificity of 0.99 is implausible.

Also, other medical statements in the text are misleading (e.g. line 25: AMD, diabetic retinopathy usually do not affect the optic disk), provided in a wrong context (cup-to-disc-ratio, not optic disc ratio), simplified too much (“Damage to the optic nerve occurs due to increased intraocular pressure.”) or simply wrong (“The optic disc gathers stimuli received through the visual nerve, which converts light stimuli into signals that can be understood, and transmits them to the brain.”)

Technical Aspects:

-              Which photo device was used for the fundus photos?

-              “The segmentation model used for retinal vessel extraction in this paper is ResU-Net”: What does this mean? Did you design the network and name it ResU-Net, or did you use a predefined net (then insert citation!)

Overall, the results are hard to interpret because of the shortcomings listed above, but generally look interesting and good.

Comments on the Quality of English Language

-

Author Response

This lacking medical understanding is first of all that glaucoma is presented as a diagnosis that can be performed on a fundus photography image (lines 213, 229). This is simply wrong. Diagnosis of glaucoma usually includes measurement of intraocular pressure, visual field testing, OCT, and funduscopy, and more often that more than one examination for a time course and determination of progression is needed. Fundus photography could be used to differentiate between glaucoma-suspect vs. non-glaucoma-suspect optic disks, or maybe to differentiate between healthy vs. late-stage-glaucoma with severe optic disk impairment, but not generally between healthy vs. glaucoma.
→ Thank you for your feedback. We plan to discuss how to address these issues with an ophthalmologist and incorporate those insights into future research.

Therefore, first of all a better description of the ODIR dataset is needed. How was the diagnosis glaucoma confirmed, and what were the criteria for healthy patients? Without further description, distinguishing between healthy vs. glaucoma solely based on fundus photography with a sensitivity/specificity of 0.99 is implausible.
→ We have supplemented the ODIR description with "Annotations are labeled by trained human readers with quality control management." The diagnosis of glaucoma was based on the annotations within the ODIR dataset. However, detailed information such as the criteria for healthy patients was not available from the dataset. This aspect will also be discussed with an ophthalmologist.

Also, other medical statements in the text are misleading (e.g. line 25: AMD, diabetic retinopathy usually do not affect the optic disk), provided in a wrong context (cup-to-disc-ratio, not optic disc ratio), simplified too much (“Damage to the optic nerve occurs due to increased intraocular pressure.”) or simply wrong (“The optic disc gathers stimuli received through the visual nerve, which converts light stimuli into signals that can be understood, and transmits them to the brain.”)
We have changed "optic disc ratio" to "cup-to-disc ratio."

We have changed “Damage to the optic nerve occurs due to increased intraocular pressure.” to “The increase in intraocular pressure due to these characteristics can also lead to damage to the optic nerve.”

The statement “The optic disc gathers stimuli received through the visual nerve, which converts light stimuli into signals that can be understood, and transmits them to the brain.” has been revised to “Visual cells convert light stimuli into signals that can be understood, and these received stimuli are transmitted to numerous ganglion cells branching from the visual nerve. These ganglion cells then gather these stimuli and send them to the optic disc. From the optic disc, neural transmission proceeds to the brain.” for further clarification.

Which photo device was used for the fundus photos?

→ We have added the following to the description of the ODIR dataset: "In these institutions, fundus images are captured by various cameras available in the market, such as Canon, Zeiss, and Kowa, resulting in varied image resolutions."

- “The segmentation model used for retinal vessel extraction in this paper is ResU-Net”: What does this mean? Did you design the network and name it ResU-Net, or did you use a predefined net (then insert citation!)

→ We used a predefined network, and have added the citation: "Diakogiannis, Foivos I., et al. 'ResUNet-a: A deep learning framework for semantic segmentation of remotely sensed data.' ISPRS Journal of Photogrammetry and Remote Sensing 162 (2020): 94-114."

Reviewer 4 Report

Comments and Suggestions for Authors

Decision: Accepted with comments to address.

Overall, the paper presents an innovative and novel approach on a multi-input CNN model incorporating Attention mechanisms for glaucoma classification. The paper details a thorough preprocessing methodology, including background adjustment and CLAHE for contrast enhancement, ensuring high-quality input data for the model. The idea and concept of the study is practically relevant to the field and the authors have demonstrated a good understanding on the topic with adequate literature review done. Robust metrics like F-1 Score, sensitivity is used to evaluate the accuracy and performance of the model and comparison to the existing models have also been done. The paper is well structured and written in a comprehensive and understandable manner. However, it is advised to incorporate the following comments.

1. Line 37-38 must be citated.

2. It is advised to include a summary of the structure of your paper at the end of the introduction section.

3. Most contents in the methodology section needs reference. Please check thoroughly and provide a reference.

4. The dataset used consists of only 32 samples for each label after preprocessing, which might not be sufficient to generalize the model’s effectiveness across diverse populations. Have you considered the effect of this to your study?

5. The preprocessing step reduces image resolution to 64x64. It is advised to study the effect of this reduction to the model since this might affect the model.

6. It is advised to address other eye diseases as well since only considering glaucoma, might limit the model's overall diagnostic utility.

7. The results show incredibly accurate and precise values (0.99,1.00). Have you considered the possibility that the data might be overfitted?

Comments on the Quality of English Language

Moderate editing of English language required for introduction, conclusion section.

Author Response

1. Line 37-38 must be cited.
→ The relevant references have been added to that section.

2. It is advised to include a summary of the structure of your paper at the end of the introduction section.
→ A summary of the paper structure has been added at the end of the introduction.

3. Most contents in the methodology section need references. Please check thoroughly and provide references.
→ References to ResU-Net and Hough Circle Transform have been added."

4. The dataset used consists of only 32 samples for each label after preprocessing, which might not be sufficient to generalize the model’s effectiveness across diverse populations. Have you considered the effect of this to your study?

→ Due to the limitations of computer specifications and the type of dataset, we had to inevitably limit the number of samples for our experiments. In the future, we plan to integrate various datasets for our studies.

5. The preprocessing step reduces image resolution to 64x64. It is advised to study the effect of this reduction to the model since this might affect the model.

→ Thank you for the suggestion. We will consider conducting experiments with different resolutions in future studies.

6. It is advised to address other eye diseases as well since only considering glaucoma, might limit the model's overall diagnostic utility.

→ Thank you for the suggestion. We will incorporate this into future research to enable classification of other eye diseases as well.

7. The results show incredibly accurate and precise values (0.99,1.00). Have you considered the possibility that the data might be overfitted?

→ Since the images used for training and validation were augmented from completely different original datasets, we judged that the probability of overfitting was low. We plan to further investigate this

Round 2

Reviewer 1 Report

Comments and Suggestions for Authors

The ROC curve is strange. Why do you have two curves? One is glaucoma and one is normal? Please explain.

Author Response

The ROC curve is strange. Why do you have two curves? One is glaucoma and one is normal? Please explain.

It is correct to have two curves, one for glaucoma and one for normal. In a classification problem like this, separate ROC curves for glaucoma and normal help us evaluate the model's performance in distinguishing between the two conditions. Each curve represents the model's ability to correctly identify that particular class (glaucoma or normal) across different thresholds.

Reviewer 3 Report

Comments and Suggestions for Authors

The authors overall performed a well job in the revision of the manuscript.

Minor points:

- In the first sentences in the introduction, it still says that AMD and diabetic retinopathy cause harm to the optic disc. Please remove this.

-  Annotations are labeled by trained human readers with quality control management --> I would change this to "the dataset also provided annotations that were labeled by trained human readers with quality control management"

Author Response

In the first sentences in the introduction, it still says that AMD and diabetic retinopathy cause harm to the optic disc. Please remove this.

→ I have removed the content related to AMD and diabetic retinopathy causing harm to the optic disc from the first sentences in the introduction as requested.

Annotations are labeled by trained human readers with quality control management --> I would change this to "the dataset also provided annotations that were labeled by trained human readers with quality control management"

→ I have updated the sentence as requested. It now reads: "The dataset also provided annotations that were labeled by trained human readers with quality control management."